# Opportunities and Challenges for Construction Health and Safety Technologies under the COVID-19 Pandemic in Chinese Construction Projects

**DOI:** 10.3390/ijerph182413038

**Published:** 2021-12-10

**Authors:** Yang Yang, Albert P. C. Chan, Ming Shan, Ran Gao, Fengyu Bao, Sainan Lyu, Qingwen Zhang, Junfeng Guan

**Affiliations:** 1The Shenzhen Research Institute, The Hong Kong Polytechnic University, Shenzhen 518057, China; albert.chan@polyu.edu.hk; 2Department of Building and Real Estate, The Hong Kong Polytechnic University, Hong Kong 999077, China; qingwen.zhang@polyu.edu.hk (Q.Z.); junfeng.guan@polyu.edu.hk (J.G.); 3Department of Engineering Management, School of Civil Engineering, Central South University, 68 South Shaoshan Road, Changsha 410004, China; ming.shan@csu.edu.cn; 4Department of Construction Economics and Management, School of Management Science and Engineering, Central University of Finance and Economics, Beijing 102206, China; ran.gao@cufe.edu.cn; 5Department of Construction Management, School of Architecture and Civil Engineering, Chengdu University, Chengdu 610000, China; baofengyu@cdu.edu.cn; 6Department of Structural Engineering, College of Civil Engineering, Hefei University of Technology, Hefei 230009, China; sainan.lyu@hfut.edu.cn

**Keywords:** COVID-19, H&S technologies, construction projects, pandemic management, innovation adoption

## Abstract

The Coronavirus disease 2019 (COVID-19) pandemic has resulted in significant delays and cost overrun in construction projects. The implementation of health and safety (H&S) technologies is one of the most important strategies to alleviate the adverse impacts of COVID-19 on the construction industry and help the industry adapt to the new normal. This study aims to evaluate the adoption of H&S technologies for pandemic management in the construction sector under the COVID-19 pandemic. Semi-structured interviews with eighteen practitioners engaged from construction companies and technology firms were conducted to collect their views on the driving forces and issues of the adoption of H&S technologies for pandemic management in Chinese construction projects. The results reveal that the major H&S technologies used included the health quick response (QR) code system, artificial intelligence (AI)-powered fever monitoring, and site access control system. These technologies were reported to be effective in preventing the spread of the pandemic in workplaces. The findings of the study amplify that the pandemic may serve as an acceleration of the adoption of H&S technologies in the construction sector. Other technologies, such as building information modeling, drones, AI-based safety monitoring, and robotics, however, were seldom used in the studied projects. The interviewees addressed several problems regarding the implementation of these technologies. High costs of technologies, a lack of client support, and disruptions to the normal work process were the main hurdles of the adoption of these technologies. The results indicated that the external influence factor—the COVID-19 pandemic—could considerably drive the use of H&S technologies, whereas the internal influence factors—cost and compatibility of technology—might be the major barriers to technology adoption. To encourage the wider use of H&S technologies in construction, the government is recommended to support the technology transformation by granting financial subsidies for costs involved in innovation adoption. Project owners may consider investing substantially in H&S technologies that can strengthen their resilient and innovative ability to adapt to the post-COVID-19 landscape. The present results will be useful to industry stakeholders and researchers interested in developing H&S technologies for combating the COVID-19 pandemic and future crises.

## 1. Introduction

Various innovative technologies have been developed to improve the performance of construction safety and promote the health and wellness of construction personnel. Despite this rapid development, the construction sector has been criticized because it is slow to adopt innovation and does not take full advantage of health and safety (H&S) technologies [1]. Nevertheless, H&S technologies might no longer be an option but a necessity because of the outbreak of coronavirus disease 2019 (COVID-19). The COVID-19 pandemic has resulted in the tremendous loss of lives, economic recession, slack businesses, high unemployment rates, and changes in the ways people live, work, and socialize [2]. Its impacts on many industries such as construction are profound. The construction sector has encountered significant difficulties with the shutdown of sites, disruptions to the supply chain, and substantial project delays [3,4,5]. To curb the spread of the pandemic, many businesses shut down at the early period, and people had to work from home. However, construction sites could not be fully closed because many projects, such as healthcare facilities, had to be continued and completed on time to protect the health and safety of the community at large [6]. Given that the virus has become a new health and safety risk on construction sites, the industry has been forced to respond to these challenges in order to achieve rapid recovery from the COVID-19 fallout.

The implementation of H&S precautionary measures is one of the most paramount strategies to curb the spread of COVID-19 in workplaces and protect the workforce [7]. H&S technologies, such as artificial intelligence (AI)-powered fever monitoring and social distance tracking, have been implemented to monitor the H&S status of construction workers at their workplaces. The COVID-19 pandemic has forced many construction companies to dramatically increase technology adoption on their projects to prevent the spread of the virus and keep projects on track [8]. In view of this, the pandemic may serve to accelerate the adoption of construction technologies [8]. However, the use of technologies to combat the pandemic may induce new challenges such as privacy, ethics, and the digital divide [9]. This concern motivated this research to explore the opportunities and issues associated with the implementation of H&S technologies in construction projects. In view of this, two major research questions are formulated: (1) Does the COVID-19 pandemic accelerate the adoption of H&S technologies on construction sites? and (2) What are the benefits and challenges with the adoption of H&S technologies under the pandemic?

Several research gaps have been identified. Many studies have investigated the impacts of COVID-19 on the construction industry in Ghana [3], the U.S. [4], the United Arab Emirates [5], Nigeria [10], and the U.K. [11]. An in-depth investigation of the impacts of the COVID-19 pandemic on Chinese construction projects is lacking. Prior research on pandemic management focused mainly on general “dos and don’ts” H&S measures [7,12,13] or “fourth industrial revolution” technologies [10]. The role of H&S technologies in managing the pandemic on site and the issues associated with technology adoption has not been well documented, especially in Chinese construction projects.

Therefore, this study aims to evaluate the adoption of H&S technologies for H&S planning and management in Chinese construction projects under the COVID-19 pandemic. The research objective of this study is twofold. First, the study investigates the impacts of COVID-19 on Chinese construction projects and identifies the typical preventive measures adopted to combat the pandemic. Second, it examines the driving forces and problems associated with the implementation of H&S technologies used to control the COVID-19 pandemic. The research findings will provide insights into how H&S technologies can effectively manage the current and future crises and promote the health and wellness of construction personnel in the post-COVID-19 era. The study will offer recommendations for building up a healthy and resilient ecosystem to facilitate the growth of H&S technology businesses and promote the wider application of H&S technologies in the construction sector.

To answer the aforementioned research questions and achieve the research objectives, the remainder of this paper is organized as follows. First, a literature review is conducted to offer an overview of the impacts of COVID-19 on the construction industry. The major H&S technologies used in construction are then introduced. A brief review of the diffusion of innovation (DOI) theory is also carried out to facilitate a better understanding of factors affecting innovation diffusion and adoption. Second, a qualitative research method is applied to gain insights into the opportunities, benefits, and challenges associated with the adoption of H&S technologies in Chinese construction projects. Lastly, the conclusions, recommendations, contributions, and limitations of the study are addressed.

## 2. Literature Review

### 2.1. Impacts of COVID-19 on the Construction Industry

There has been a profound impact of COVID-19 on the construction industry through direct and indirect ways [3]. Construction workers reported a higher number of COVID-19 cases than other occupational workers [14,15], indicating a high risk of infection on construction sites. This is probably because the labor-intensive nature of construction poses challenges to the feasibility of social distancing in workplaces [16]. The COVID-19 crisis led to the temporary closure of construction sites and interrupted the supply chain across the globe. The pandemic caused a delay in most ongoing construction projects and reduced the productivity of the construction industry, resulting in significant cost and time overrun [3,10,17,18]. Indirectly, the pandemic led to significant economic losses and high unemployment rates [10,19]. Other indirect impacts of COVID-19 on the construction industry included reduced profits, lowered investment, and reduced wages of staff [17,20].

Health and safety guidelines have been recommended for site activities to minimize the spread of the virus and enable construction sites to return to normal [21]. The Occupational Safety and Health Administration (OSHA) [22] has formulated measures to manage the pandemic on construction sites. The health screening process shall be implemented before workers enter the site to minimize health risks. Managerial measures have also been formulated by developing policies, procedures, and preparedness plans for workplace controls [22]. Following the standard guidelines, the major anti-epidemic measures implemented on construction sites included workforce education, provision of personal hygiene products, and regular health checks [3]. A set of good practices have been implemented to prevent the spread of the virus on construction sites, including maintaining social distancing [6]. Ayat et al. [23] summarized five categories of COVID-19 mitigation measures. Construction technologies, including prefabrication, building information modeling (BIM), and digital tools, have been implemented to alleviate the impact of the pandemic on the construction sector [23].

The pandemic, however, is perceived to bring some positive outcomes. These outcomes include an increase in technology adoption, cleaner sites, closer collaboration between project stakeholders, and a shift toward a more resilient supply chain [6,13]. The COVID-19 pandemic has accelerated the technology transformation in the construction industry, stimulated investment in technologies [24], and facilitated the adoption of advanced digital technologies [25]. The technologies used on construction sites under COVID-19 have also improved communication among project stakeholders and thus enhanced project efficiency and productivity [13,26]. Rapid deployment of advanced digital technologies might help prevent and mitigate the adverse impacts of a similar pandemic in the future [27]. It is worthwhile to examine how emerging technologies are reshaping construction activities under the pandemic; however, relevant academic literature has been scant [10].

### 2.2. Overview of H&S Technologies in Construction

H&S technologies have received mounting attention in the construction industry [28,29,30]. Technologies, such as virtual and augmented reality (VR and AR), BIM, unmanned aerial vehicles (UAVs), wearable equipment, and internet of things (IoT) technologies, have been developed and applied to prevent construction accidents and improve site health and safety [1]. The application of visualization technologies such as BIM, VR, and AR enables users to detect health and safety-related risks [1,29,31,32,33]. IoT technologies have been applied to prevent collisions between equipment and workers by detecting the real-time distance between them [34]. Smart bracelets, information and communication technologies, and artificial intelligence (AI) algorithms have been leveraged to develop an early warning system that measures the risk of heat strain of construction workers [35]. AI-powered image recognition can help detect whether construction workers wear personal protective equipment properly [36]. Construction automation and robotics have been employed for autonomous installation and operation to minimize workers’ exposure to safety risks [37,38].

The use of H&S technologies could help reduce, mitigate, or control the risks of the pandemic [39,40]. There are many H&S technologies used on construction sites under the pandemic. AI-powered fever monitoring devices are useful for health screening. AI-powered image recognition can help detect the compliance of social distancing and mask wearing of construction workers. The deployment of remote-control and semi-autonomous construction equipment through the 5G network reduces the reliance on onsite labor and thus minimizes the exposure to health risks. The use of BIM and other digital platforms enables the project team to seamlessly communicate and coordinate when social distancing measures are implemented.

Previous studies have identified the driving forces and constraints to the adoption of diverse technologies in the construction industry [10,41]. More specifically, these technologies can improve project performance concerning cost–time performance, productivity, and safety performance. These benefits are the major driving forces of the wide adoption of technologies. The government favoring policy and regulations and market needs also significantly drive the use of H&S technologies in construction. By contrast, high initial costs of hardware and software, high costs of technology implementation, and a lack of skilled experts are some critical barriers to the adoption of technologies. Moreover, poor policy support, poor information and communication technology (ICT) infrastructure, and conservative culture may impede the use of technologies in the construction sector.

### 2.3. Diffusion of Innovation (DOI) Theory

Diffusion of innovation is “the process by which an innovation is communicated through certain channels over time among the members of a social system” [42]. The DOI theory implies that the diffusion of innovation in an organization can be mainly driven by internal and external influence factors [43]. External influences are those that exist outside of the adopting organization such as policies, regulations, advertisement, market demand, competitive advantage, or demonstrable benefits of innovation [43,44,45,46,47]. Regulations can force companies to adopt innovation through detailed specifications and standard procedures and encourage innovations through incentives, subsidies, and grants [45,48]. The DOI theory has been widely used in the research of ICT to provide insight into the adoption, implementation, infusion, and diffusion of ICT innovations [5]. Without strong internal or external motivations, an organization will not evolve to use innovations. A construction organization’s internal and external environment may considerably influence the investment decision and innovation adoption [49]. The DOI theory implies that internal and external factors can drive or impede innovation adoption. The internal factors are found more significant in driving the adoption of innovations in the construction sector than external factors [43,44,50]. The DOI theory may help explain the underlying factors affecting the adoption of H&S technologies in this study.

## 3. Research Method

Qualitative research can be employed to analyze complex social phenomena in natural settings by exploring the views and experiences of participants [51]. This approach is suitable when researchers have limited knowledge about new phenomena [51,52,53,54]. Hence, the qualitative research method was applied to gain fresh information about the phenomenon and facts of the studied topic. This approach offers an opportunity in dealing with the study context about the impacts of COVID-19 on Chinese construction projects and the adoption of H&S technologies. Previous research on technology adoption on construction sites during the COVID-19 pandemic [10] guided the research method of this study. The interview survey has been widely used as an important data collection method in qualitative research [55]. The semi-structured interview was carried out by using a standard interview guide that lists out the open-ended questions to collect the primary data (Appendix A). This approach was adopted because it could facilitate an interactive discussion on the standardized interview questions [56]. A purposive interviewee selection approach was adopted to ensure the reliability and quality of the interview survey. First, interviewees from different professionals and organizational backgrounds were invited, such as owners, contractors, and technology firms. Second, the interviewees with adequate work experience (i.e., 3 years and above) were invited to the survey. About 24 interview invitations were emailed to the potential respondents, of which 18 were received back and used for analysis. The sample size of the interview survey is considered adequate since similar surveys invited nine to twelve [3,10]. Accordingly, 18 interviews from 18 companies participated in the survey from late July 2021 and September 2021. The participant’s position, year of experience, location of the primary project, and their organization’s background are given in Table 1. The survey focused on investigating the early and concurrent impacts of the COVID-19 pandemic in Chinese construction projects. The study was approved by the PolyU Institutional Review Board (Reference Number: HSEARS20211029006).

A quantitative and qualitative content analysis of the interview data was performed. Content analysis is a commonly used approach in analyzing data gathered by interviews or focus groups [57]. It is suitable for answering questions such as: “what are the concerns of people about a topic?” and “what reasons do people have for using or not using a service or procedure?” [58]. Content analysis can be applied to either qualify or quantify data by grasping trends, patterns, frequency, and structures of textual information [57]. This method was applied in this study to identify the most critical issues concerned by interviewees. The process of the content analysis generally followed that of Stemn et al. [59]. First, the interview scripts for each question were divided into meaning units. These units were then condensed by extracting the core meaning and refining the context. Second, sorting the condensed meaning units into similar or related codes as descriptive labels was performed. Third, codes were divided into categories and/or subcategories. For instance, anti-epidemic measures could be divided into three categories: personal, managerial, and technological measures. A category or subcategory consists of codes that appear to deal with the same issue. After grouping data into categories, the frequency of each coded item was determined to address the critical issues concerned by the interviewees. The interviews with the Hong Kong practitioners were conducted in English, while others were carried out in Chinese. The Chinese scripts were translated into English by the researchers for further data analysis.

## 4. Results and Discussion

The interviewees provided relevant information based on their views of the impacts of COVID-19 on their construction projects, the adoption of H&S technologies under the pandemic, and the driving forces and issues associated with the implementation of H&S technologies.

### 4.1. Impacts of the COVID-19 Pandemic on the Chinese Construction Projects

The COVID-19 rules and regulations prevent the spread of the virus by restricting the movement of people and crowd gathering [10]. The impact of lockdown resulted in the temporary closure of construction sites, restricted site activities, labor shortage, and disruption to the supply chain. As a consequence, there was a delay in most construction projects, and some projects had to be postponed. This consequence was addressed by 80% of interviewees. The delay, in turn, increased construction costs due to work extension. The rising cost was the major consequence of the pandemic since the expenses on material, labor, transportation and logistics, machinery and administration, and extra expenses on anti-epidemic measures escalated the construction cost. The time and cost overrun further resulted in the reduced investment turnover and financial losses of the companies. Participant 17N pointed out that “although the rising cost was caused by force majeure, it is not easy to claim compensation from the owner”. This implied that the pandemic had caused negative contractual responsibilities on construction projects [11]. A reduction in investment in the new projects was identified (Participants 1A, 17N, 20O). However, some interviewees opined that the Chinese government had greatly supported infrastructure investment to speed up economic recovery from the pandemic. “Due to the comprehensive formulation and implementation of COVID-19 rules in China, the impact of COVID-19 on construction projects could be minimal”, commented Participant 2B.

Although limited, Participant 1A pointed out that the pandemic probably brought about certain positive impacts to the industry. For instance, the pandemic was perceived to accelerate the use of information technologies and digital transformation. The technologies could monitor the health of the onsite workforce and improve the efficiency of project management. This finding may help ascertain how emerging technologies are reshaping construction practices [10]. Figure 1 outlines the categories of the impacts of the COVID-19 on the studied Chinese construction projects.

### 4.2. Anti-Epidemic Measures

Figure 2 illustrates the typical anti-epidemic measures implemented in the studied Chinese construction projects. All interviewees responded that the project staff and workers had to strictly follow the COVID-19 rules and procedures on construction sites. The anti-epidemic measures could be broadly classified into three categories, personal, managerial, and technological measures. Personal controls included mask wearing, daily disinfection, vaccinations, quarantine, health screening for temperature checkups, and regular nucleic acid tests. There were several mobile specimen collection stations on site providing workers with free COVID-19 nucleic acid testing services (Participant 24R). The management team changed or developed work practices to cope with the challenges of the COVID-19 pandemic. For instance, the management staff had to work from home, reduce business trips, and hold online meetings (Participants 5E, 7G, 8H, 10I, 14L, 20O). They developed and implemented the epidemic prevention guidelines and emergency management systems on construction sites (Participants 1A, 12J, 17N, 24R). The project team adopted job rotation and arranged small-group work for laborers to prevent crowd gathering (Participant 20). They also strengthened site supervision and safety management to ensure that all staff and workers complied with the COVID-19 rules (Participants 1A, 4D, 6F, 20O). Workforce education programs were offered by the employers to enhance the implementation of epidemic preventive measures (Participants 4D, 6F, 15M). The last category is the technological approach. The construction site access control systems that consist of fever monitoring, facial recognition, and labor information were implemented for health screening purposes (Participants 4D, 20O). The health quick response (QR) code systems were used in Mainland China but not in Hong Kong or Dubai. The system recorded the personal information about their COVID-19-related health conditions, vaccination conditions, and travel history. The use of QR codes could easily monitor and track the potential contact cases and thus help prevent the spread of the virus on construction sites. However, data and privacy breaches have received grave concern regarding the deployment of the QR code system [60]. Despite these risks, technological surveillance and tracking measures have positive outcomes in combating the pandemic [60,61]. This is also confirmed by the interviewees of the study.

The H&S measures could reduce the health risk, but some affected work and management efficiency. For instance, some interviewees criticized that the implementation of anti-epidemic measures increased costs (Participants 4D, 6F, 10I, 14L) and the workload of site personnel (Participants 17N, 20O). Findings agreed with Ezeokoli et al. [62] that enforcing these guidelines to the construction sites could induce extra costs to the project and might disrupt the normal site operations. Some workers even did not strictly follow the COVID-19 rules (Participant 17N). Participant 20O indicated the work inefficiency problem due to social distancing, echoed by Zheng et al. [16]. Akinlolu et al. [1] also found that the social distancing between workers on site of around 1.5–2 m challenged the way construction activities were traditionally undertaken.

### 4.3. The Adoption of H&S Technologies in Construction Projects during and in the Post Pandemic

Most of the interviewees agreed that the outbreak of COVID-19 accelerated the application of H&S technologies to combat the COVID-19 pandemic. These technologies are used to curb the spread of the virus in order to meet the mandatory requirements of the government on public health protection. “The application of relevant technologies could reduce the risks and losses caused by COVID-19 to the project and improve the efficiency of the implementation of relevant COVID-19 preventive measures”, as addressed by Participant 4D. This finding shows the benefits of the application of H&S technologies. The proven benefits, in turn, would facilitate the wider use of H&S technologies on construction sites. Beyond the construction industry, “COVID-19 has accelerated the development of the Internet, robotics, logistics, and AI industries” (Participant 12J). However, Participants 2B, 20O, and 21P indicated that “technologies were merely the supporting tools for pandemic prevention” because the pandemic management heavily relied on personal and managerial measures.

One of the most widely used H&S technologies on site was the health QR code system (80%). It tracks the individual’s travel history and personal health status by displaying a real-time barcode in one of three colors representing different risk levels [63]. However, the health QR code system is only prevalent in Mainland China but not in Hong Kong or Dubai. AI-powered fever monitoring devices (100%) and the construction site access control system (85%) had been widely used on construction sites. The “COVID-19 big data analysis map” (10%) was applied, enabling the management to identify medium- and high-risk areas and formulate corresponding coping strategies for people from those risk areas. These technologies assisted in H&S management during the pandemic. Automatic disinfection equipment, unmanned cleaning vehicles, smart helmets, digital office platforms, location tracking mobile apps, and AI-powered cameras were implemented in some projects. Previous literature and market review identified that drones, BIM technology, and automatic equipment have been developed and applied as anti-epidemic measures on construction sites. However, these technologies were absent in this survey.

Most of the interviewees opined that the financial burden was one of the major hurdles making H&S technologies unfavorable in construction projects. COVID-19 resulted in cost overrun and extra expenses on anti-epidemic measures. This situation made the project team reluctant to spend extra costs on H&S technologies. The compliance to COVID-19 rules by health screening, vaccination, regular nucleic acid tests, and maintaining social distancing was deemed adequate to prevent the spread of the virus and keep projects on track. Some interviewees questioned the cost–benefit of the adoption of various H&S technologies. Participant 20O commented: “…each technology has its limitations. For example, the temperature in Dubai is extremely high. We have to complete body temperature measurement before 7:30 every day; otherwise, the detected body temperature will be more than 40 Celsius degree caused by a high ambient temperature”. The interviewee further added that “…using AI cameras to monitor social distance. Many factors could make the technology impractical. If used, AI cameras are required to be installed on each floor, which highly relies on the internet network. Additional staffing should be arranged to check the detection results. And raising alerts to non-compliant workers requires extra efforts of supervisors. Some places, such as staff canteen, elevator, and washroom, are not suitable to install AI cameras. And what if AI cameras were damaged by someone…” When asking why not use drones in their project, Participant 20O further mentioned: “my project is a high-rise building project, so it is difficult to use drones indoors”. And “BIM technology is seldom used in Dubai or Mainland China because of a lack of BIM integration across the whole supply chain (owner, consultant, builder)”. All these issues limited the use of H&S technologies in construction projects. The underlying problem could lie in the poor compatibility of these technologies to diverse, dynamic, and complex construction environments, as summarized by Participants 20O and 23Q. In addition, Participant 7G pointed out that “resistance to change” and “a lack of motivation” were some hurdles to the adoption of H&S technologies in construction projects.

In terms of the change in investment in technology before and after COVID-19, 60% of interviewees observed that there was a minimal change or no change in the investment in H&S technologies. However, extra expenses were spent on personal anti-epidemic measures. “If the owner does not require the use of technology, the construction company will not actively increase the investment in technologies”, addressed by Participant 20O. Only 30% indicated that the investment in H&S technologies increased 5% or above. Among them, the technology company (Participant 23Q) indicated that their investment in H&S technologies increased double. The remaining 10% of participants had no idea. Most of the interviewees showed a positive attitude to the development of H&S technologies in the future and pointed out that drones, automated equipment, offsite construction, and AI technologies would have profound impacts on improving the performance of construction projects. However, prior to the spread of H&S technologies in construction, it would be important to develop relevant supporting policy and industry standards, addressed by Participants 13K and 24R.

The present study attempts to apply the DOI theory to explain the underlying reasons why some H&S technologies were widely used under the pandemic, but others were not. The DOI theory has introduced various internal and external factors driving or impeding the adoption of innovation. Previous studies indicated that the internal factors are more significant in driving the adoption of innovations in the construction sector than external factors [43,44,50]. This argument is partially supported by the current study. On the one hand, the proven benefits of using H&S technologies in combating COVID-19 promoted innovation adoption on construction sites. Hence, the COVID-19 pandemic, as an external factor, could considerably accelerate the adoption of H&S technologies in the construction industry. On the other hand, some construction practitioners lack motivation in using H&S technologies due to extra costs and efforts. As a consequence, the H&S technologies used have been limited to the health QR code system, AI-powered fever monitoring, and site access control system, while BIM, drones, AI-powered video surveillance, and robotics were not widely used in the studied projects. The findings imply that internal influence factors (e.g., costs, compatibility of technology) are the major constraints of innovation adoption. Findings are consistent with previous studies on innovation adoption [10,64]. The study also found that the COVID-19 pandemic and the governmental regulation enforcement considerably facilitated the applications of the health QR code system, AI-powered fever monitoring, and site access control system. The finding helps explain why the COVID-19 pandemic, as an external factor, could accelerate the adoption of H&S technologies in the construction industry. Since the use of these H&S technologies has adequately complied with COVID-19 rules, some construction practitioners lack motivation in using other technologies associated with extra costs and efforts. In this regard, internal influence factors might be the major hurdles of innovation adoption. Therefore, the findings echoed the DOI theory that internal and external factors might drive or impede the adoption of H&S technologies in construction. More specifically, external factors could be significant in driving the adoption of H&S technologies in the construction sector, while internal factors could be the predominant constraints of technology adoption.

## 5. Conclusions

The study highlights the impacts of COVID-19 on Chinese construction projects and the issues affecting the use of H&S technologies under the pandemic. The reported adverse effects of the pandemic included remarkable delays on projects, difficulty in securing labor and materials, and increased construction costs. The widely implemented anti-epidemic measures could be classified into personal, managerial, and technological measures. Technological tools such as online meetings and health screening had been somewhat efficient in preventing the spread of pandemics and maintaining normal work procedures. The major H&S technologies used included the health QR code system, AI-powered fever monitoring, and construction site access control system. They were forced on construction sites in order to meet the mandatory requirements of government authorities on public health protection. Other technologies, such as BIM, drones, AI-powered video surveillance, and robotics, were seldom used in the studied projects. The interviewees addressed several problems regarding their implementation on site. High costs, a lack of client support, and disruptions to the normal work process were some major hurdles of the adoption of these technologies. The findings implied that the external influence factor—the COVID-19 pandemic—could promote the use of H&S technologies, whereas the internal influence factors—cost and compatibility of technology—might inhibit technology adoption.

The findings offer several implications for accelerating the rate of diffusion of H&S technologies in the construction sector. The government plays an important role in promoting the wide use of H&S technologies on construction sites. They are recommended to continue to support construction companies interested in technology transformation. For instance, financial subsidy for costs involved in innovation adoption (e.g., Construction Innovation and Technology Fund in Hong Kong) may help relieve the financial burden of construction companies (especially medium-/small-sized firms) and promote wider use of innovative technologies. Commitment from clients is also crucial to developing and cultivating a favoring environment for technology transformation in construction projects. Public work contracts can be prioritized to construction companies that are willing or capable to use innovative technologies. Technology firms and researchers are recommended to develop sophisticated and tailor-made technologies for the construction industry to minimize the adverse impacts of disruptive technologies on normal work procedures.

This empirical study contributes to offering an in-depth understanding of the adoption of H&S technologies under the COVID-19 pandemic by identifying the driving forces and issues associated with the adoption of H&S technologies in Chinese construction projects. To the authors’ best knowledge, it is one of the first studies addressing this matter. The study offers empirical evidence concerning external driving forces and internal organizational hurdles to the adoption of H&S technologies. The findings may guide policymakers, construction practitioners, technology firms, and researchers concerning how H&S technologies can be used as effective anti-pandemic measures in construction projects and how those technologies can be widely used in future crises.

This study is not without limitations such as potentially biased sampling. To be specific, only one overseas Chinese project was studied. The institutional environment between Mainland China and overseas is different; as a result, the health QR code system is only prevalent in Mainland China but not in Dubai. Future research can be undertaken to deepen the understanding of the studied topic by comparing the H&S technologies adoption between Mainland and overseas Chinese construction projects. The impact of the institutional environment on innovation adoption will be then addressed. Furthermore, the present qualitative research generated two hypotheses that (1) the external influence factor—the COVID-19 pandemic—could drive the use of H&S technologies, (2) the internal influence factors—cost and compatibility of technology—might impede the technology adoption. These hypotheses can later be tested by quantitative studies [51,52] to further explore the relationship between H&S technologies adoption and internal/external influence factors while taking the influence of COVID-19 into account.

## Figures and Tables

**Figure 1 ijerph-18-13038-f001:**
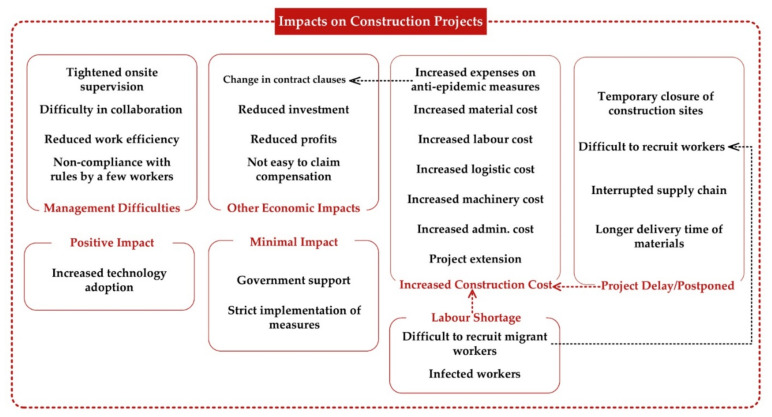
The impacts of COVID-19 on construction projects. Dotted line implies the potential causal relationship based on the interviewees’ responses.

**Figure 2 ijerph-18-13038-f002:**
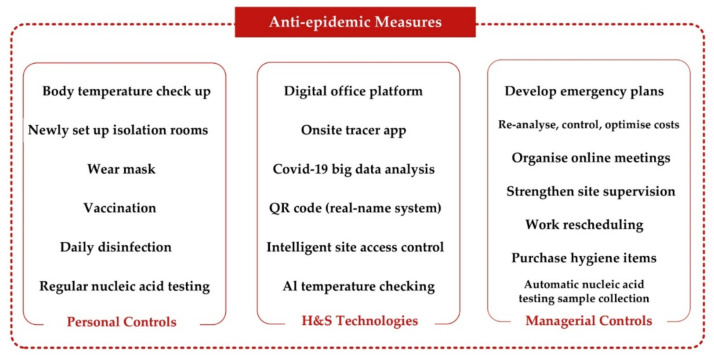
The COVID-19 preventive measures.

**Table 1 ijerph-18-13038-t001:** Background information about the interviewees.

Interviewee ID	Company	Job Position	Work Experience (Years)	Project Location
1A	Developer (private)	General Project Manager	3	Sichuan
2B	Consulting firm (design)	Business Manager	7	Changsha
3C	Developer (state-owned)	Engineer	17	Changsha
4D	Developer (state-owned)	Engineer	6	Changsha
5E	Developer (private)	Cost Manager	11	Changsha
6F	Developer (state-owned)	Project Manager, BIM Supervisor	9	Changsha
7G	Technology consultant	Managing Director	18	Hong Kong
8H	Technology consultant	Director	15	Hong Kong
10I	Developer (private)	BIM Project Manager	15	Shanghai
12J	Consulting firm (design)	Senior Engineer	14	Shanghai
13K	Consulting firm (design)	Engineer	5	Shandong
14L	Consulting firm (quantity surveying)	Quantity Surveyor	8	Changsha
15M	Contractor (joint venture)	Project Manager	6	Changsha
17N	Contractor (state-owned)	Assistant Engineer	3	Sichuan
20O	Contractor (state-owned)	Project Manager	13	Dubai
21P	Contractor (state-owned)	Manager	22	Sichuan
23Q	Technology consultant	Senior Engineer	12	Shanghai
24R	Contractor (state-owned)	Safety Engineer	8	Wuhan

## Data Availability

The interview data presented in this study are available on reasonable request from the corresponding author.

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
