# Peer review of "Opportunities and Challenges for Construction Health and Safety Technologies under the COVID-19 Pandemic in Chinese Construction Projects"

_ijerph, 2021, doi:10.3390/ijerph182413038_

Round 1
Reviewer 1 Report
This is an interesting paper which aims to evaluate the adoption of H&S technologies used for H&S planning and management in the construction sector under the Covid-19 pandemic. Semi-structured interviews with eighteen construction companies and technology firms were conducted to collect views and data on the driving forces and issues of the adoption of H&S technologies for pandemic management in Chinese construction projects. However, despite the noted strengths, the study could benefit from an improved research methods section, enhanced articulation of the theoretical contributions and better explanations of the limitations.
Research methods
Page 4, Line 176: The authors have provided explanations for using the interviews as a data collection technique used. However, the usage of ‘structured interview’ is not explained, and the rationale / justification should be extended to this as well. The question that would arise might be around why not use semi-structured OR unstructured interviews?
Conclusion section- This study is underpinned by the diffusion of innovation theory (DOI). However, the conclusion section fails to leverage this by explaining how the theory as applied could be used for articulating the ‘why’s” of the phenomena under investigation. How does the IDT as employed contribute to the theory testing or development?
Page 9, Lines 882-384: What kind of contribution is being referred to in the following sentence: “This study contributes to offering an in-depth understanding of the adoption of H&S technologies under the Covid-19 pandemic by identifying the driving forces and issues associated with the technology adoption”? Is it methodological, conceptual or empirical?
Limitations – The following statement needs further elaboration: “This study is not without limitations such as the limited sample size and biased sampling” – The key question is around the sample size as the authors had already claimed that this was adequate and included supporting studies around the appropriateness of the sample size. Therefore, how does this become an issue?
Reviewer 2 Report
Dear authors,
I have an excellent opportunity to review your research paper : Opportunities and Challenges for Construction Health and Safety Technologies under COVID-19 Pandemic in Chinese Construction Projects.
I found many interesting concepts and ideas, which were put into discussion, for example: Semi-structured interviews with eighteen construction companies and technology firms were conducted to collect views and data on the driving forces and issues of the adoption of H&S technologies for pandemic management in Chinese construction projects. Therefore, I’m sure that this topic is very interesting for publication due to current online environmental circumstances and COVID19.
There are some strong advantages of this manuscript:
- the manuscript is well-organized and has several sections;
- the topic is actual and the problem is announced: this study finds t the pandemic might accelerate the use of digital and information technologies
- manuscript includes both theoretical discussion and practical case study;
- This study identifies a few challenges that The major H&S technologies used included the health quick response (QR) code system, artificial intelligent (AI)-based facial temperature measurement, and construction site access control system;
- authors present results and give some recommendations: The interviewees addressed several problems regarding the implementation of H&S technologies onsite. High costs of technologies, a lack of client support, and disruptions to the normal work process were the main hurdles of the adoption of these technologies. The findings of the study amplify that the pandemic may serve as an acceleration of the adoption of technologies in the construction sector;
- I agree with the main thesis and believe the findings. Everything is logical, clear, and well-ordered.
There are some major recommendations, which should be improved in manuscript:
- There is special format for references, please, check it one more time; please, add some more actual references: Shvetsova, O.A., Lee, J.H. Minimizing the environmental impact of industrial production: Evidence from South Korean waste treatment investment projects/ Applied Sciences (Switzerland), 2020, 10(10), pp. 3489-3510 and Olga A. Shvetsova; Lee, S.-K. Living Labs in University-Industry Cooperation as a Part of Innovation Ecosystem: Case Study of South Korea. Sustainability Journal, 2021, 13, 5793.
-
Ayat, M., Malikah and Kang, C.W. (2021), "Effects of the COVID-19 pandemic on the construction sector: a systemized review", Engineering, Construction and Architectural Management, Vol. 9 No2 https://doi.org/10.1108/ECAM-08-2021-0704
-
S H Zamani et al 2021 IOP Conf. Ser.: Earth Environ. Sci. 682 012049
-
Alenezi N A T 2020 The Impact Of Covid-19 On Construction Projects In Kuwait, International Journal of Engineering Research and General Science 8 July-August, 2020 ISSN 2091-2730
- Zhong BL, Luo W, Li HM, Zhang QQ, Liu XG, Li WT, et al. Knowledge, attitudes, and practices towards COVID-19 among Chinese residents during the rapid rise period of the COVID-19 outbreak: a quick online cross-sectional survey. Int J Biol Sci. (2020) 16:1745. doi: 10.7150/ijbs.45221
- It could be interesting to mention advantages and disadvantages of chosen methods and explain the limitations; also there should research questions be addressed; please, give the clear structure of your research (steps and explanations due to the research questions);
- Maybe better quality of graphical material (figure or table) can impress reading with visualization effect;
- You mention in text, that a lack of internal motivation for the adoption of H&S technologies in construction companies, in this case more details about problem are required (reasons, importance and so on).
Thank you one more time for this interesting contribution, I have a great pleasure to read it. Congratulations to Authors!
Round 2
Reviewer 2 Report
Thanks to authors for excellent job! All recommendations were completed successfully. Just check one more time writing (typing) mistakes
Author Response
Thank you for the reviewer's valuable comments on our manuscript. The comments significantly help improve the readability of the article. Please, we tried to correct typos throughout the manuscript. Below are our corrections.
Line 19: delay --> delays
Line 28: intelligent --> intelligence
Line 29: jobsites --> workplaces
Line 35: external --> the external
Line 36: internal --> the internal
Line 39: subsidy --> subsidies
Line 83: the pandemic --> pandemic
Line 86: have --> has
Line 95: health --> the health
Line 101: reminder --> remainder
Line 137: of --> in; jobsites --> sites
Line 168: the health --> health
Line 173: with respect to --> concerning
Line 194: the internal --> internal
Line 201: views --> the views
Line 216: in --> to
Line 226: on --> of
Line 230: applied either qualifying or quantifying --> applied to either qualify or quantify
Line 269: commented by --> commented
Line 289: practice --> practices
Line 301: system-->systems
Line 305: helped--> help
Line 310: some of them --> some
Line 322: in post--> in the post
Line 329: indicate-->shows
Line 330: the wider --> wider
Line 336: individual's-->the individual's
Line 346: drone--> drones
Line 355: concerned about -->questioned
Line 361: internet --> the internet
Line 366: drone-->drones
Line 369: a -->the
Line 397: lack of-->lack
Line 409: lack of --> lack
Line 415: predominate --> predominant
Line 431: external-->the external
Line 432: internal-->the internal
Line 440: financial-->the financial
Line 442: develop-->developing
Line 443: cultivate-->cultivating
Line 444: which-->that
Line 447: the normal-->normal
Line 453: intra-->internal
Line 457: the future-->future
Line 458: potential-->potentially
Line 464: may-->will
Line 465: external--> the external
Line 466: internal--> the internal